# Cognitive impairment and reporting of hypertension among adults in india: Evidence from a population-based study

Priyanka Dixit[1]*, Basil Edolikkandy[2], Montu Bose[1], Waquar Ahmed[1], Shiva Halli[3]

**1** School of Health Systems Studies, Tata Institute of Social Sciences, Mumbai, India, **2** Department of Public Health and Mortality Studies, International Institute for Population Sciences, Mumbai, India, **3** Department of Community Health Sciences, Faculty of Medicine, University of Manitoba, Manitoba, Canada

* priyanka.dixit@tiss.ac.in

## Abstract

The study investigates the disparities between the prevalence of self-reported and measured hypertension among adults and the role of cognitive impairment in such disparities. The study used data from the first wave of the Longitudinal Aging Study in India, a nationally representative survey of 72,250 individuals. Percentage distributions were calculated for cognitive impairment, self-reported hypertension, and objective measures of hypertension along with the explanatory variables. Multivariable logistic regression was performed to assess the association of cognitive impairment and other factors with self-reported and measured hypertension. Furthermore, the Propensity Score Matching method was used to estimate the effect of cognitive impairment on self-reported hypertension, measured hypertension, and the misreporting of hypertension. Cognitive impairment was found in 9.8% of Indian adults in this study. Cognitive impairment was most prevalent among females and those over 75 years of age. Hypertension too was higher among females as well as among rural residents and those with no education compared to their respective counterparts. The likelihood of cognitively impaired adults having hypertension was 24% more than that of their cognitively unimpaired counterparts [OR = 1.24, CI = 1.18-1.32]. Other risk factors of hypertension were age, alcohol consumption, and place of residence. The PSM analysis revealed that individuals with cognitive impairment were 2.7% more likely to underreport their hypertensive status compared to those without cognitive impairment. The study underscores the significance of acknowledging reporting bias among individuals with cognitive impairment. Addressing this bias in healthcare systems is crucial. The policy recommendations encompass creating tailored healthcare interventions, improving access to healthcare, enhancing communication strategies, and providing robust support to those with cognitive impairment to ensure accurate diagnosis and proper disease management. Healthcare providers require training to identify and mitigate reporting biases in this vulnerable group. Doing so will ultimately enhance healthcare outcomes.

**Data availability statement:** The study uses secondary data which is available upon request through https://www.iipsindia.ac.in/content/lasi-wave-i. The data are also available in the repository of the Gateway to Global Aging Data (https://g2aging.org/). This is to confirm that others researchers would be able to access these data in the same manner as the authors and that the authors did not have any special access privileges that others would not have.

**Funding:** The authors received no specific funding for this work.

**Competing interests:** The authors have declared that no competing interests exist.

## Introduction

Hypertension accounts for 19% of all non-communicable disease (NCD) deaths worldwide [1]. According to the World Health Organization (WHO), out of the total income loss of INR 1094–1113 billion arising from NCDs in India, a loss of INR 199 billion was on account of hypertension [2] alone. Numerous population-based studies from low-income countries have shown that various socio-demographic characteristics are associated with hypertension, with higher rates of the disease being consistently associated with older age, female sex, and lower levels of education [3,4]. According to specific epidemiological data, hypertension is also linked to behavioural risk factors such as smoking, drinking, insufficient physical activity, and poor eating behaviours [3–5]. The data provided by large-scale surveys on the prevalence of hypertension is usually based on self-reports and standard tests. The inherent bias in the self-reporting of illnesses [6–8] contributes to inaccuracies in disease reporting, which are further exacerbated by conditions such as cognitive impairment.

Cognitive impairment occurs due to the normal ageing process for various reasons and can affect the quality of life and even the ability to live independently [5,9,10]. It often affects elderly persons, adding to their functional disabilities and creating dependence, while also posing challenges in adapting to the changes in health technology. Meanwhile, young onset cases are also increasingly becoming common [11–14]. Evidence from the developed and developing nations suggests that age-related morbidities, functional limitations, poor mental health, and low quality of life among older adults are all associated with poor cognitive health [10,12,15–17]. Adults suffering from cognitive impairment may experience memory loss and face issues with understanding, concentrating, and decision-making, all of which can lead to the false reporting of diseases, including hypertension, during the collection of data on self-reported morbidity [18,19].

The Longitudinal Aging Study in India, Wave 1 (LASI)[20], provides self-reported and measured hypertension data of persons aged 45 years and above. Self-reported health indicators from patients are becoming increasingly acceptable in research. However, given the significant prevalence of cognitive issues in this population, any questionnaire for evaluating cognitive impairment must be reliable and valid, especially when utilizing such measures over an extended period [21]. Previous research has examined the discrepancy between self-reported hypertension and objective evaluations of hypertension and between false-negative and false-positive reporting factors and found the prevalence of self-reported and measured hypertension among older adults to be 27.4% and 31.5%, respectively, and the prevalence of total hypertension (average SBP ≥ 140 mmHg or/and DBP ≥ 90 mmHg or current use of any antihypertensive medication) to be 42.3%[22]. This significant difference between self-reports and the actual disease burden can impact the public health policies in the country.

The retrospective nature of a self-reporting question means that the answer to it relies on the recollection of past events, which highlights the role of episodic memory or the ability to recall such events. These cognitive functions hold significant importance. Notably, the earliest signs of Alzheimer's disease often manifest as a decline

in episodic memory [6]. In the context of this study, the role of cognitive function is pivotal in the reporting of health conditions. Therefore, the study aims to expand the existing knowledge regarding the impact of cognitive impairment on the self-reported and clinically measured prevalence of hypertension in the adult population of India. It also provides insights into the other socio-economic and health-related variables associated with the prevalence of self-reported and measured hypertension and the misreporting of hypertension among adults.

## Materials and methods

### Data

Data was drawn from the first wave of the Longitudinal Aging Study in India (LASI), a nationally representative survey of 72,250 adults aged 45 years and above (including spouses irrespective of their age) conducted across all states and UTs of India, except Sikkim. LASI focuses on the scientific investigation of the health, economic, and social determinants of population ageing as well as its consequences in India. LASI is a partnership between the International Institute for Population Sciences (IIPS), Harvard T. H. Chan School of Public Health (HSPH), and the University of Southern California under the direction of the Ministry of Health and Family Welfare (MoHFW), Government of India. LASI, Wave 1, adopted a multistage stratified area probability cluster sampling design, including a three-stage sampling design in rural areas and a four-stage sampling design in urban areas. Further details on the methodology used by the LASI survey can be found in the LASI survey report [20].

### Outcome variable

The self-reported data on hypertension was obtained from the question "*Has any health professional ever told you that you have hypertension or high blood pressure?*" The participant was identified as hypertensive if s/he answered "Yes." Systolic and diastolic measurements were taken three times on an electronic monitor (Omron model H7121) with a one-minute gap between each reading in a sitting position. Based on the protocol recommended by the WHO, the average of the last two readings of the systolic blood pressure (SBP) and the diastolic blood pressure (DBP) were taken into consideration. In keeping with the guidelines of the Seventh Joint National Committee on Detection, Evaluation, and Treatment of Hypertension (JNC-VII), an individual was considered to be hypertensive if s/he had a systolic blood pressure (SBP) of at least 140 mmHg and a diastolic blood pressure (DBP) of at least 90 mmHg or if s/he used any antihypertensive medication [23].

### Explanatory variables

The primary independent variable was cognitive impairment. It was calculated across five domains (memory, orientation, arithmetic function, executive function, and object naming) derived from the Health and Retirement Study (HRS) cognition module. A composite cognitive index (0–43) was constructed by combining these five domains, and the participants in the lowest ten percentile were used as a proxy measure of poor cognitive functioning [20,24]. The description of each individual domain, along with scoring criteria for each domain is provided in the S1 Table.

   The rest of the explanatory variables were grouped based on their individual, behavioural, household, and community characteristics. The individual factors were: Age in completed years (<45, 45–54, 55–64, 65–74, 75+); Sex (Male, Female), Educational level (No education, Primary, Secondary, Higher); Working status (Never worked, Currently working, Not presently working); Marital status (Currently married, Widowed, Other); and Body Mass Index (BMI) (Normal, Underweight, Overweight). The behavioural factors were: Physical activity level (Inactive, Active), Smoking status (Never, Former, Current), and Alcohol use (No, Yes). The household factors were: Religion (Hindu, Muslim, Christian, Other), Caste (Scheduled caste, Scheduled tribe, Other backward class (OBC), Other), and Monthly Per Capita Expenditure (MPCE) class (Poorest, Poorer, Middle, Richer, Richest). Finally, the community-level factors were: Place of residence (Rural, Urban) and Region (North, Central, East, Northeast, West, South).

## Statistical analysis

LASI provides information on both self-reported and objective measures of hypertension. We began by calculating the percentage distribution of adults in terms of cognitive impairment and non-cognitive impairment along with the background variables. Next, we constructed a frequency table to show the distribution of self-reported and measured hypertension with the explanatory variables. We used a multivariable logistic regression model [25] to assess the association between cognitive impairment with self-reported and measured hypertension after adjusting the effect of various individual- (age, sex, education, working status, marital status, and BMI), behavioural- (smoking status, physical activity, and alcohol consumption), household- (MPCE quintile, religion, and caste), and community-level factors (place of residence and region). We presented the results as odds ratios (ORs) with 95% confidence intervals (CI). In order to estimate the actual effect of cognitive impairment on hypertension reporting, we used Propensity Score Matching.

Out of the total 72,250 respondents, 9,574 with missing data on cognitive function were excluded from both the multivariable and PSM analyses, resulting in a final analytical sample of 62,676 respondents. Furthermore, due to the lack of suitable matches, 35 observations in Model 1, 46 in Model 2, and 285 in Model 3 were identified as off-support and subsequently excluded from the PSM analysis., Appropriate sampling weights were used in the analyses to obtain the estimates. All the analyses were performed using STATA 17 [26].

## About propensity score matching

We used the propensity score matching (PSM) method to make a comparison of the average outcome for those with cognitive impairment and those without it. The PSM method allowed us to compare the outcome between the treated and the controlled observations given various background characteristics [27]. It is given by the following equation:

$$p(X) = Pr(D = 1|X)$$

Where $X$ is a multidimensional vector of pre-treatment characteristics, and $D = \{0, 1\}$ is the indicator variable of exposure to cognitive impairment.

The impact of cognitive impairment for an individual, $i$, denoted by $\delta_i$ is the difference between the potential outcome in the presence of cognitive impairment ($Y_1$) and the potential outcome in the absence of cognitive impairment ($Y_0$).

$$\delta_i = Y_1 - Y_0$$

The average impact of cognitive impairment on the outcome is provided by the Average Treatment Effect (ATE). This is given by:

$$ATE = E(\delta_i) = E(Y_1 - Y_0)$$

A counterfactual model was constructed to calculate the average effect of cognitive impairment on the treated individuals. Using this model, the Average Treatment Effect on the Treated (ATT) was calculated as a measure of the average increase in the outcome of those who were cognitively impaired compared to those who weren't cognitively impaired and had similar characteristics after matching.

$$ATT = E(Y_1|D = 1) - E(Y_0|D = 1)$$

Where $E(Y_1|D = 1)$ is the average outcome of the treated individuals, and $E(Y_0|D = 1)$ is the average outcome the treated individuals would have obtained in the absence of treatment.

Like ATT, the Average Treatment Effect on the Untreated (ATU) too was calculated as a measure of the average increase in the outcome of those who were not cognitively impaired as compared to those who were cognitively impaired and had similar characteristics after matching.

$$ATU = E\left(Y_1\middle|D = 0\right) - E(Y_0|D = 0)$$

Where $E\left(Y_1\middle|D = 0\right)$ is the average outcome of the untreated individuals, and $E(Y_0|D = 0)$ is the average outcome the untreated individuals would have obtained in the presence of treatment.

Three PSM models were constructed to assess the effect of cognitive impairment on hypertension. Model 1 estimated the effect of cognitive impairment on self-reported hypertension. Model 2 evaluated the effect of cognitive impairment on measured hypertension. Model 3 assessed the role of cognitive impairment in the misreporting of hypertension (defined as diagnosed hypertensive cases among those who did not self-report hypertension). The standard errors of average treatment effects were calculated using the bootstrapping method with 100 replications [28]. All the explanatory variables considered in the multivariate logistic regression model for the PSM model were matched. This was done because matching based on many variables increases the likelihood of the propensity score matching assumption holding.

### Ethics approval and consent to participate

The study is based on secondary data from a survey that was approved by the "Indian Council of Medical Research (ICMR) Ethics Committee in January 2017 and, for which, prior written or oral informed consent was obtained from the participants." All methods were carried out in accordance with the relevant guidelines and regulations and in accordance with the World Medical Association Declaration of Helsinki.

### Results

Fig 1 exhibits the percentage of cognitive impairment and self-reported and measured hypertension among adults. It can be observed that 9.8% of all adults in India had cognitive impairment. The prevalence of self-reported hypertension was 26.4%, whereas that of measured hypertension was 30.2%.

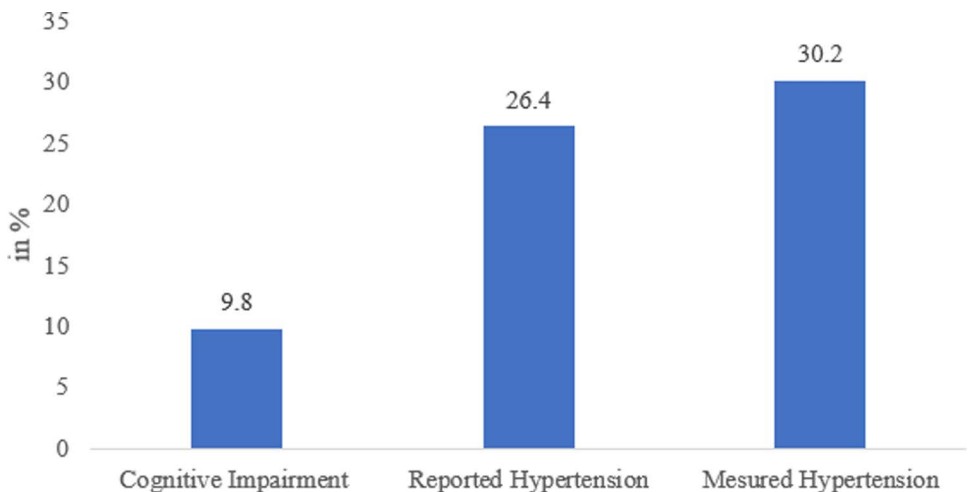

**Fig 1. Percentage of cognitive impairment and self-reported and measured hypertension among adults.** Source: Authors' computation based on LASI data, 2017-18.

Table 1 presents the weighted distribution of the background characteristics of the total sample of cognitively impaired and non-cognitive impaired (NCI) adults. More than one-fourth (25.6%) of the respondents with cognitive impairment were above 75 years of age. Females constitute 78% of the cognitively impaired population, while males account for 22%. Nearly, 91.2% of adults with cognitive impairment had no education compared to 45.7% of those without cognitive impairment. Among adults having cognitive impairment, about 54.2% were currently married and about 52.6% were from the poorest and poorer households. About 74.7% of adults having cognitive impairment were physically inactive compared with 67.4% of those without cognitive impairment. As many as 86.5% of adults with cognitive impairment had never smoked and 85.46% had never consumed alcohol. Nearly 83% of adults having cognitive impairment lived in rural areas, while this proportion was lower (66.9%) in the case of those without cognitive impairment.

Among NCI individuals, approximately one-third (34.63%) were aged 45–54 years, while only 7.57% were aged 75 years and above. A higher proportion of NCI were females (56.03%). Almost 49.12% of NCI were currently working. The majority of NCI adults had never attended school (44.04%). Among individuals with NCI, 79.21% were currently married. Physical inactivity was reported among 65.74% of NCI adults. In terms of substance use, 84.17% of NCI individuals had never smoked, and 85.84% had never consumed alcohol. Also, a lower proportion (66.44%) of NCI adults resided in rural areas compared to those with cognitive impairment (82.89%).

The effect of cognitive impairment on self-reported and measured hypertension is presented in Table 2. While 33.8% of cognitively impaired adults had hypertension, only 25.41% reported it. The proportion of self-reported (34.8%) and measured (38.33%) hypertension was higher among those aged above 75 years in comparison to < 64 years. Compared to females, males were found to have a wider difference in self-reported prevalence (23.4%) and measured hypertension (31.7%). The prevalence of hypertension was higher among adults with higher levels of education (Self-reported: 30.3%; Measured: 33.1%) in comparison to primary and no education individuals. Also, the proportion of hypertension was higher among adults who were former smokers (Self-reported: 31.8%, Measured: 33.7%) in comparison to current and never smokers. While the proportion of self-reported hypertensive individuals was higher among those who did not consume alcohol compared to those who never consumed alcohol (27.3% compared to 21.2%), the proportion of those with measured hypertension was higher in the case of those who consumed alcohol (29.3% compared to 35.7%). Adults living in urban areas had a higher prevalence of hypertension (Self-reported: 35.8%; Measured: 34.3%) than their rural counterparts.

Table 3 shows the effects of various factors on hypertension reporting using the binary logistic regression model. While adults having cognitive impairment had a higher risk of having hypertension [OR: 1.24, CI: 1.18-1.32], their likelihood of reporting it was lower [OR: 0.89, CI: 0.84-0.95]. The effect of cognitive impairment on self-reported and measured hypertension did not change significantly upon introducing control variables to the model, which shows that cognitive impairment had an individual effect on hypertension. The probability of self-reported and measured hypertension went up as age increased. Compared to males, females had a 29% higher rate of hypertension reporting [AOR: 1.29, CI: 1.22-1.36]. MPCE had a significant effect on hypertension reporting but not on measured hypertension. Adults with higher BMI (Overweight) had a higher likelihood of self-reporting hypertension than those with lower BMI (Normal and underweight); the same pattern was visible in the case of measured hypertension. Compared to adults who did not consume alcohol, those who did had a 43% higher risk of measured hypertension [AOR: 1.43 CI: 1.36-1.51]. Place of residence, too, had a significant effect on self-reported and measured hypertension. Urban residents had a 25% higher rate of self-reported hypertension and a 5% higher rate of measured hypertension than their rural counterparts.

The estimated propensity matching scores of cognitive impairment along with the outcome variables are given in Table 4. The ATT estimates in Model 1 show that without matching, adults with cognitive impairment were, on average, 3.1% less likely to report hypertension than those without cognitive impairment. The results also indicate that after matching with various background variables, the difference between the treatment and the control groups was 3.6%, signifying that if a person had cognitive impairment, s/he had a 3.6% less chance of reporting that s/he was hypertensive compared

**Table 1. Characteristics of Cognitively impaired and Non-Cognitive impaired Adults in India, LASI 2017-18.**

| Background Variables | Cognitive impaired | | Non-Cognitive impaired | |
|---|---|---|---|---|
| | Sample size | Percentage | Sample size | Percentage |
| **Age** | | | | |
| <45 years | 261 | 4.13 | 5751 | 9.47 |
| 45-54 years | 1016 | 16.62 | 20533 | 34.63 |
| 55-64 years | 1474 | 24.4 | 16378 | 27.81 |
| 65-74 years | 1756 | 29.69 | 10893 | 20.52 |
| 75+years | 1515 | 25.16 | 3866 | 7.57 |
| **Sex** | | | | |
| Male | 1365 | 21.99 | 25430 | 43.97 |
| Female | 4657 | 78.01 | 31991 | 56.03 |
| **Educational level** | | | | |
| No education | 5327 | 91.15 | 23165 | 44.04 |
| Primary | 597 | 7.85 | 15220 | 5.39 |
| Secondary | 85 | 0.88 | 12450 | 18.86 |
| Higher | 13 | 0.11 | 6585 | 11.71 |
| **Working status** | | | | |
| Never worked | 2063 | 30.19 | 16431 | 26.87 |
| Currently working | 1920 | 31.73 | 27704 | 49.12 |
| Not currently working | 2038 | 38.08 | 13283 | 24.02 |
| **Marital status** | | | | |
| Currently married | 3280 | 54.2 | 45968 | 79.21 |
| Widowed | 2555 | 43.47 | 9771 | 18.24 |
| Other | 187 | 2.33 | 1680 | 2.54 |
| **MPCE quintile** | | | | |
| Poorest | 1796 | 28.26 | 10443 | 19.76 |
| Poorer | 1364 | 24.35 | 11306 | 20.74 |
| Middle | 1159 | 20.42 | 11609 | 20.58 |
| Richer | 945 | 15.5 | 12073 | 20.23 |
| Richest | 758 | 11.46 | 11990 | 18.69 |
| **BMI** | | | | |
| Normal | 2760 | 49.71 | 27528 | 51.11 |
| Underweight | 1837 | 37.44 | 8372 | 18.72 |
| Overweight | 741 | 12.85 | 17018 | 30.18 |
| **Physical activity level** | | | | |
| Inactive | 4550 | 74.65 | 38568 | 65.74 |
| Active | 1456 | 25.35 | 18761 | 34.26 |
| **Smoking status** | | | | |
| Never | 5126 | 86.46 | 47577 | 84.17 |
| Former | 223 | 3.75 | 2396 | 3.39 |
| Current | 659 | 9.79 | 7355 | 12.45 |
| **Alcohol consumption** | | | | |
| No | 5042 | 85.46 | 47653 | 85.84 |
| Yes | 966 | 14.54 | 9689 | 14.16 |
| **Religion** | | | | |
| Hindu | 4235 | 80.29 | 41888 | 81.95 |
| Muslim | 722 | 10.98 | 6992 | 11.65 |

*(Continued)*

**Table 1.** (Continued)

| Background Variables | Cognitive impaired | | Non-Cognitive impaired | |
| --- | --- | --- | --- | --- |
| | Sample size | Percentage | Sample size | Percentage |
| Christian | 788 | 4.83 | 5760 | 2.84 |
| Other | 277 | 3.9 | 2778 | 3.56 |
| **Caste** | | | | |
| Scheduled caste | 1157 | 25.71 | 9374 | 19.28 |
| Scheduled tribe | 1902 | 18.32 | 9299 | 7.58 |
| OBC | 1771 | 39.01 | 21674 | 46.75 |
| Other | 937 | 16.97 | 14910 | 26.38 |
| **Place of residence** | | | | |
| Rural | 4955 | 82.89 | 35715 | 66.44 |
| Urban | 1067 | 17.11 | 21706 | 33.56 |
| **Region** | | | | |
| North | 950 | 12.25 | 10302 | 11.95 |
| Central | 719 | 16.53 | 7117 | 18.76 |
| East | 1267 | 29.52 | 10247 | 24.42 |
| Northeast | 1007 | 4.65 | 8013 | 3.97 |
| West | 992 | 18.35 | 7744 | 16.80 |
| South | 1087 | 18.69 | 13998 | 24.08 |

Source: Authors' computation based on LASI data, 2017–18.

to those who were not cognitively impaired. After controlling for different background variables through propensity score matching, the findings of the Average Treatment Effect on the Treated (ATT) analysis showed a 3.6% decrease in the likelihood of individuals with cognitive impairment reporting hypertension compared to those without cognitive impairment.

In Model 2, the ATT values for the treatment and the control groups were 0.364 and 0.360, indicating that if cognitive impairment was not present among adults with cognitive impairment, the prevalence of hypertension would have been 0.4% lower (the difference between the ATT of the treatment and the control groups). The ATU values for the treatment and the control groups were 0.315 and 0.324, respectively. This suggests that if the adults without cognitive impairment developed impairment, their likelihood of having hypertension would have increased by 0.8%. The difference in ATE, which shows the average treatment effect, was 0.8%.

Model 3 considered those adults who reported being non-hypertensive but were diagnosed as being hypertensive. The ATT values for the treatment and the control groups were 0.245 and 0.218, respectively. This implies that individuals with cognitive impairment exhibited a 2.7% higher probability of underreporting their hypertensive condition compared to those without cognitive impairment. This also indicates that among those with cognitive impairment, only 21.8% would have inaccurately indicated hypertension if they were not impaired, whereas this figure was 24.5% for individuals with cognitive impairment. The ATU values show that if the adults who did not have cognitive impairment had impairment, their chance of misreporting hypertension condition would have decreased by 0.7%. The ATE value shows that the difference in misreporting between the treated and untreated adults was 0.4%.

## Discussion

Ageing is associated with various geriatric syndromes. The frequency of sickness and disability, particularly cognitive impairment, rises with age [29]. Cognitive impairment is characterized by memory loss, learning disabilities, and lack of perceivability and concentration [30]. A systematic review of previous studies shows that the global prevalence of cognitive

**Table 2. Prevalence of Self-Reported and Measured Hypertension by Cognitive Impairment and Other Background Characteristics in India, LASI 2017-18.**

| Background Variables | Self-reported hypertension | | Measured hypertension | |
|---|---|---|---|---|
| | Sample size | Percentage | Sample size | Percentage |
| **Cognitive impairment** | | | | |
| No | 18377 | 26.52 | 19107 | 29.89 |
| Yes | 1540 | 25.41 | 2014 | 33.83 |
| **Age** | | | | |
| <45 years | 1028 | 14.87 | 1024 | 17.10 |
| 45-54 years | 5100 | 20.60 | 6042 | 25.67 |
| 55-64 years | 5887 | 27.63 | 6336 | 31.15 |
| 65-74 years | 5328 | 34.11 | 5183 | 37.21 |
| 75+years | 2534 | 34.81 | 2536 | 38.33 |
| **Sex** | | | | |
| Male | 7535 | 23.40 | 9589 | 31.67 |
| Female | 12342 | 28.60 | 11532 | 29.17 |
| **Educational level** | | | | |
| No education | 8343 | 23.80 | 9467 | 29.41 |
| Primary | 5114 | 27.70 | 5379 | 30.30 |
| Secondary | 4066 | 30.05 | 4197 | 30.77 |
| Higher | 2354 | 30.25 | 2078 | 33.14 |
| **Working status** | | | | |
| Never worked | 7125 | 33.08 | 6185 | 29.73 |
| Currently working | 6349 | 18.24 | 8872 | 27.76 |
| Not currently working | 6402 | 33.93 | 6062 | 35.26 |
| **Marital status** | | | | |
| Currently married | 14043 | 24.30 | 15136 | 28.11 |
| Widowed | 5256 | 34.55 | 5311 | 37.77 |
| Other | 577 | 20.55 | 673 | 29.30 |
| **MPCE quintile** | | | | |
| Poorest | 2956 | 20.20 | 4080 | 29.36 |
| Poorer | 3551 | 23.51 | 4183 | 29.60 |
| Middle | 4006 | 25.86 | 4310 | 30.67 |
| Richer | 4485 | 29.45 | 4303 | 30.22 |
| Richest | 4879 | 34.33 | 4245 | 31.44 |
| **BMI** | | | | |
| Normal | 8170 | 23.15 | 10378 | 28.90 |
| Underweight | 1867 | 15.21 | 2812 | 21.67 |
| Overweight | 7771 | 38.95 | 7658 | 38.43 |
| **Physical activity level** | | | | |
| Inactive | 15194 | 29.56 | 14983 | 31.29 |
| Active | 4555 | 19.73 | 6105 | 27.96 |
| **Smoking status** | | | | |
| Never | 16927 | 27.31 | 17575 | 30.43 |
| Former | 971 | 31.78 | 995 | 33.71 |
| Current | 1850 | 18.60 | 2515 | 27.79 |
| **Alcohol consumption** | | | | |
| No | 17095 | 27.26 | 16960 | 29.34 |

*(Continued)*

**Table 2.** (Continued)

| Background Variables | Self-reported hypertension | | Measured hypertension | |
|---|---|---|---|---|
| | Sample size | Percentage | Sample size | Percentage |
| Yes | 2664 | 21.20 | 4133 | 35.65 |
| **Religion** | | | | |
| Hindu | 13921 | 25.12 | 14808 | 29.61 |
| Muslim | 2931 | 33.50 | 2712 | 31.95 |
| Christian | 1801 | 25.94 | 2408 | 31.69 |
| Other | 1224 | 34.18 | 1192 | 37.79 |
| **Caste** | | | | |
| Scheduled caste | 3073 | 23.66 | 3368 | 28.45 |
| Scheduled tribe | 2424 | 15.00 | 4178 | 30.41 |
| OBC | 7436 | 26.72 | 7497 | 30.18 |
| Others | 6058 | 31.24 | 5282 | 31.23 |
| **Place of residence** | | | | |
| Rural | 10919 | 22.09 | 13068 | 28.43 |
| Urban | 8958 | 35.78 | 8053 | 34.30 |
| **Region** | | | | |
| North | 4439 | 33.55 | 3873 | 31.20 |
| Central | 1847 | 19.59 | 2317 | 25.04 |
| East | 3141 | 24.56 | 3465 | 28.24 |
| Northeast | 2266 | 27.84 | 3023 | 33.10 |
| West | 2688 | 26.73 | 2782 | 34.10 |
| South | 5496 | 29.96 | 5661 | 33.15 |
| **India** | 19877 | 26.42 | 21121 | 30.22 |

Source: Authors' computation based on LASI data, 2017–18.

impairment ranges from 5.1% to 41%, with a median of 19%[30]. The first wave of LASI collected data on measured cognition across five distinct domains. The mean score in each of the five cognitive domains was lower among the elderly (60 and above) compared to the adults (45 and above) in India [20]. In this study, we found that 9.8% of adults had cognitive impairment in India. Earlier studies show that the prevalence of cognitive impairment among the elderly is between 3.5% and 11.5% in India [5,31].

This study found that the prevalence of cognitive impairment was significantly higher among adults living in rural areas, which is consistent with the findings of other studies conducted in different countries around the globe [32–34]. Additionally, persons with no education had a higher percentage of cognitive impairment than those who were literate. The higher prevalence of cognitive impairment in rural areas may be primarily due to lower educational attainment in those areas. This is supported by the fact that rural areas have lower educational attainment than urban areas [35] and that education is negatively correlated with the prevalence of cognitive impairment in rural areas [32]. Low educational attainment has been linked to low-skill jobs in less cognitively stimulating environments, which have been reported to further increase the risk of cognitive impairment [36,37].

The prevalence of self-reported hypertension among adults was 26.4%, while the prevalence of measured hypertension in the same population was nearly 31%. To develop effective local and national strategies for preventing and managing hypertension, it's vital to understand the actual prevalence of the condition in the population. While self-reported health data is commonly regarded as a reliable indicator of population health in the developed nations, its accuracy is less studied and uncertain in lower-income countries. Since clinical diagnosis of hypertension through surveys can be expensive and time-consuming,

**Table 3. Multivariable logistic regression of the relationship between self-reported and measured hypertension with socioeconomic characteristics in India, LASI 2017-18.**

| Background variables | Self-reported hypertension | | Measured hypertension | |
|---|---|---|---|---|
| | Unadjusted OR (95% CI) | Adjusted OR (95% CI) | Unadjusted OR (95% CI) | Adjusted OR (95% CI) |
| **Cognitive Impairment** | | | | |
| No | Ref. | Ref. | Ref. | Ref. |
| Yes | 0.89*** (0.84-0.95) | 0.84*** (0.78-0.91) | 1.24*** (1.18-1.32) | 1.15*** (1.04-1.23) |
| **Age** | | | | |
| <45 years | | Ref. | | Ref. |
| 45-54 years | | 1.82*** (1.67-1.98) | | 1.74*** (1.61-1.89) |
| 55-64 years | | 2.84***(2.61-3.09) | | 2.47*** (2.28-2.68) |
| 65-74 years | | 3.9*** (3.56-5.27) | | 2.98*** (2.73-3.25) |
| 75+years | | 4.2*** (3.78-4.67) | | 3.48*** (3.14-3.85) |
| **Sex** | | | | |
| Male | | Ref. | | Ref. |
| Female | | 1.29*** (1.22-1.36) | | 0.82*** (0.78-0.86) |
| **Educational level** | | | | |
| No education | | Ref. | | Ref. |
| Primary | | 1.19*** (1.13-1.25) | | 1.04* (0.99-1.09) |
| Secondary | | 1.12*** (1.06-1.19) | | 1.07** (1.02-1.13) |
| Higher | | 1.16*** (1.08-1.25) | | 1.01 (0.94-1.09) |
| **Working status** | | | | |
| Never worked | | Ref. | | Ref. |
| Currently working | | 0.75*** (0.71-0.79) | | 0.91** (0.87-0.96) |
| Not currently working | | 1.13*** (1.07-1.19) | | 1 (0.95-1.05) |
| **Marital status** | | | | |
| Currently married | | Ref. | | Ref. |
| Widowed | | 1.19*** (1.13-1.25) | | 1.35*** (1.29-1.42) |
| Other | | 0.95 (0.84-1.06) | | 1.14** (1.03-1.26) |
| **MPCE quintile** | | | | |
| Poorest | | Ref. | | Ref. |
| Poorer | | 1.15*** (1.08-1.22) | | 0.99 (0.93-1.04) |
| Middle | | 1.29*** (1.21-1.37) | | 1 (0.94-1.05) |
| Richer | | 1.4*** (1.32-1.49) | | 0.96 (0.91-1.02) |
| Richest | | 1.52*** (1.42-1.62) | | 0.93* (0.87-0.98) |
| **BMI** | | | | |
| Normal | | Ref. | | Ref. |
| Underweight | | 0.59*** (0.56-0.63) | | 0.64*** (0.61-0.68) |
| Overweight | | 1.85*** (1.78-1.94) | | 1.6*** (1.54-1.67) |
| **Physical activity level** | | | | |
| Inactive | | Ref. | | Ref. |
| Active | | 0.88*** (0.84-0.92) | | 0.98 (0.94-1.02) |
| **Smoking status** | | | | |
| Never | | Ref. | | Ref. |
| Former | | 1.13* (1.03-1.24) | | 0.84*** (0.77-0.92) |
| Current | | 0.87*** (0.81-0.93) | | 0.83*** (0.78-0.88) |
| **Alcohol consumption** | | | | |

*(Continued)*

**Table 3.** (Continued)

| Background variables | Self-reported hypertension | | Measured hypertension | |
|---|---|---|---|---|
| | Unadjusted OR (95% CI) | Adjusted OR (95% CI) | Unadjusted OR (95% CI) | Adjusted OR (95% CI) |
| No | | Ref. | | Ref. |
| Yes | | 1.01 (0.95-1.08) | | 1.43*** (1.36-1.51) |
| **Religion** | | | | |
| Hindu | | Ref. | | Ref. |
| Muslim | | 1.32*** (1.25-1.41) | | 1.17*** (1.10-1.24) |
| Christian | | 1.05 (0.97-1.14) | | 0.96 (0.89-1.04) |
| Other | | 1.29**(1.18-1.41) | | 1.24*** (1.14-1.35) |
| **Caste** | | | | |
| Scheduled caste | | Ref. | | Ref. |
| Scheduled tribe | | 0.72*** (0.67-0.78) | | 1.28*** (1.20-1.37) |
| OBC | | 0.97 (0.91-1.02) | | 0.91** (0.86-0.96) |
| Others | | 1.04 (0.98-1.10) | | 0.99 (0.93-1.05) |
| **Place of residence** | | | | |
| Rural | | Ref. | | Ref. |
| Urban | | 1.25*** (1.20-1.31) | | 1.05** (1.01-1.10) |
| **Region** | | | | |
| North | | Ref. | | Ref. |
| Central | | 0.66*** (0.62-0.71) | | 0.88*** (0.82-0.94) |
| East | | 0.86*** (0.80-0.92) | | 0.97 (0.91-1.03) |
| Northeast | | 0.91* (0.84-0.99) | | 1.11** (11.03-1.20) |
| West | | 0.79*** (0.74-0.85) | | 0.99 (0.92-1.05) |
| South | | 0.93* (0.88-0.99) | | 1.17*** (1.10-1.24) |

*, **, *** refer to <0.05, <0.01, and <0.001 levels of significance respectively.

Source: Authors' computation based on LASI data, 2017–18.

information is often gathered through self-reporting. Many studies have used self-reported data to explore how hypertension affects various health-related factors, including quality of life, mental health, chronic conditions, and even mortality.

This study attempted to explore whether cognitive impairment impacts an individual's capacity to effectively recall and communicate health information. Cognitive impairment may potentially result in a lower propensity to report disorders. The credibility of self-reported disease measures, particularly in adults with cognitive decline, is a topic of concern. Despite a lack of existing literature on the impact of cognitive impairment on the accurate self-reporting of hypertension, this study attempted to investigate this relationship among adults. The self-reported hypertension variable was created from the question "*Has any health professional ever told you that you have hypertension or high blood pressure?*" The answer was subject to the respondent's ability to recall and to his/her acceptability to be identified as a patient. By comparing self-reported hypertension with the measured data, the study aimed to substantiate the hypothesis that cognitive impairment contributes to the underreporting of hypertension.

The multivariate logistic regression analysis shows that the risk of hypertension among adults with cognitive impairment was higher than their counterparts with no cognitive impairment. At the same time, the risk of self-reported hypertension was lower among adults with cognitive impairment. This variation between the odds ratios of self-reported and measured hypertension shows that cognitive impairment plays an important role in the reporting of hypertension, which may also be true in the case of other chronic conditions. Further research is warranted to gain a better understanding and provide more clarity on this matter.

**Table 4. Measurement of Effects of Cognitive Impairment on Reporting and Objective Diagnosis of Hypertension Through Propensity Score Matching in India, LASI 2017-18.**

|  |  | Sample | Treated | Controls | Difference | S.E. | T-stat |
|---|---|---|---|---|---|---|---|
| **Self-reported hypertension** | **Model 1** |  |  |  |  |  |  |
|  |  | Unmatched | 0.241 | 0.273 | -0.031 | 0.006 | -4.87 |
|  |  | ATT | 0.241 | 0.278 | -0.036 | 0.005 | 0.31 |
|  |  | ATU | 0.273 | 0.296 | 0.022 |  |  |
|  |  | ATE |  |  | 0.018 |  |  |
| **Measured hypertension** | **Model 2** |  |  |  |  |  |  |
|  |  | Unmatched | 0.364 | 0.315 | 0.049 | 0.006 | 7.21 |
|  |  | ATT | 0.364 | 0.360 | 0.004 | 0.001 | 0.39 |
|  |  | ATU | 0.315 | 0.324 | 0.008 |  |  |
|  |  | ATE |  |  | 0.008 |  |  |
| **Misreporting of hypertension** | **Model 3** |  |  |  |  |  |  |
|  |  | Unmatched | 0.245 | 0.194 | 0.051 | 0.005 | 8.83 |
|  |  | ATT | 0.245 | 0.218 | 0.027 | 0.009 | 2.82 |
|  |  | ATU | 0.194 | 0.187 | -0.007 |  |  |
|  |  | ATE |  |  | -0.004 |  |  |

Source: Authors' computation based on LASI data, 2017–18.

The results of the PSM also indicated that the adults who did not have cognitive impairment had a 3.6% less chance of self-reporting hypertension as compared to those who had cognitive impairment. In contrast, the underreporting of hypertension was 2.7% more likely among cognitively impaired adults as compared to those who did not have cognitive impairment. These results explain, to a great extent, the reason for the enormous dissimilarity between the data on self-reported and measured hypertension [6,22,38]. This raises concerns and establishes the necessity for further research to understand the area more deeply and combine tools on self-reporting of hypertension and cognitive impairment.

This study has both strengths and weaknesses. The strengths include the use of a larger and more representative sample size of elderly individuals in India and the addition to the existing literature on the effect of cognitive impairment on self-reported and measured hypertension. As far as the limitations are concerned, we could not illustrate the precise long-term impact of cognitive impairment among the elderly due to the lack of data availability. The data from the LASI follow-up waves will allow us to assess the amount of cognitive decline and its impact on the elderly in India.

## Conclusion

The significant impact of cognitive impairment on hypertension reporting explains the reduced validity of self-reporting measures in reporting a disease among adults. Addressing this issue requires going beyond merely narrowing the awareness gap to taking care of the shortcomings of the healthcare system and, as such, necessitates thorough monitoring of the adult population and provision of informed guidance to them due to the potential elevated risk and prevalence of various diseases among them compared to self-reported data. Screening of adults for cognitive impairment and establishment of geriatric departments at all levels of healthcare should be undertaken since cognitive impairment greatly affects the quality of life of the aged and their caretakers. Earlier and better interventions would aid in detecting the disease and its treatment, resulting in a higher quality of life for everyone.

## Supporting information

**S1 Table. A description of domain-wise cognitive measures.**
(DOCX)

## Acknowledgments

The Longitudinal Aging Study in India Project is funded by the Ministry of Health and Family Welfare, Government of India, the National Institute on Aging (R01 AG042778, R01 AG030153), and the United Nations Population Fund, India.

## Author contributions

**Conceptualization:** Priyanka Dixit, Montu Bose.

**Data curation:** Priyanka Dixit, Basil Edolikkandy, Waquar Ahmed.

**Formal analysis:** Priyanka Dixit, Basil Edolikkandy, Waquar Ahmed.

**Investigation:** Priyanka Dixit.

**Methodology:** Priyanka Dixit, Waquar Ahmed.

**Supervision:** Priyanka Dixit, Shiva Halli.

**Validation:** Priyanka Dixit.

**Writing – original draft:** Priyanka Dixit, Basil Edolikkandy.

**Writing – review & editing:** Priyanka Dixit, Montu Bose, Waquar Ahmed, Shiva Halli.

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
