## [Decision Letter · Decision Letter 0]

21 Feb 2024

PMEN-D-24-00001

Cognitive Impairment among the Adults & Reporting of Hypertension in India: Evidence from a Population-based Study

PLOS Mental Health

Dear Dr. Priyanka Dixit

Thank you for submitting your manuscript to PLOS Mental Health. After careful consideration, we feel that it has merit but does not fully meet PLOS Mental Health’s publication criteria as it currently stands. Therefore, we invite you to submit a revised version of the manuscript that addresses the points raised during the review process.

We look forward to receiving your revised manuscript.

Kind regards,

Barkın Köse

Academic Editor

PLOS Mental Health

Journal Requirements:

1. Please amend your online Financial Disclosure statement. If you did not receive any funding for this study, please simply state: “The authors received no specific funding for this work.”

2. Please update your online Competing Interests statement. If you have no competing interests to declare, please state: “The authors have declared that no competing interests exist.”

3. Please provide separate figure files in .tif or .eps format only and remove any figures embedded in your manuscript file. Please also ensure that all files are under our size limit of 10MB. You may leave the figure captions or legends in the manuscript.

For more information about how to convert your figure files please see our guidelines: https://journals.plos.org/mentalhealth/s/figures

Additional Editor Comments (if provided):

Reviewers' comments:

Reviewer's Responses to Questions

**Comments to the Author**

1. Does this manuscript meet PLOS Mental Health’s publication criteria ? Is the manuscript technically sound, and do the data support the conclusions? The manuscript must describe methodologically and ethically rigorous research with conclusions that are appropriately drawn based on the data presented.

Reviewer #1: Partly

Reviewer #2: No

2. Has the statistical analysis been performed appropriately and rigorously?

Reviewer #1: Yes

Reviewer #2: No

3. Have the authors made all data underlying the findings in their manuscript fully available (please refer to the Data Availability Statement at the start of the manuscript PDF file)?

Reviewer #1: Yes

Reviewer #2: No

4. Is the manuscript presented in an intelligible fashion and written in standard English?

Reviewer #1: No

Reviewer #2: No

5. Review Comments to the Author

Reviewer #1: Dear Editor,

This proposed study was conducted to investigate whether the presence of cognitive impairment affects the presence of slef report hypertension and I think it can contribute to the literature. However, the language of the article should definitely be revised again in terms of grammar and flow. In addition, the points that I think would take the article to the next level are as follows. If the language is corrected and revisions are made to the following points, I think it can be published in your journal.

1. In line 59, too little fruit and vegetables does not explain diet-induced hypertension. The relevant sentence can be corrected as "poor eating behaviors". I think it is not correct to attribute hypertension only to fruit and vegetable consumption in terms of nutrition.

2. I can say that the Introduction is not written in a fluent and explanatory way. Hypertension was determined as the universe of the subject, information about hypertension was given first, but the connection was not established in the transition from paragraph 1 to paragraph 2. However, it would be more appropriate to bring the subject to self-report in the 1st paragraph and then mention that cognitive impairment may negatively affect this.

3. I think it is not appropriate to talk about Alzheimer's and dementia because the research does not try to define these diseases.

4. The Cognitive paragraph in the Introduction section should be rewritten in a way that is linked to the previous paragraph and in line with the purpose of the research.

5. I think that the table names and table layouts are not in accordance with the journal's spelling rules.

6. Paragraph 1 of the discussion section contains information that should be included in the introduction. It would be more accurate to prioritize the main findings of the research and explain the value of these data to the literature.

7. Discussion 323. Line; “Cognitive impairment is one of the most untreated ailments in India when compared to developed countries, and it is one of the most critical public health challenges. If left undetected in the elderly, it may progress to Dementia or Alzheimer.” This is not the purpose of the study, it is inappropriate to mention about it.

8. In line 359-362 “This variation between the odds ratios of self-reported and measured hypertension shows that cognitive impairment plays an important role in hypertension reporting, which might also be true in the case of other chronic conditions.” I think it is not correct to use such definitive sentences in the discussion section. Due to the nature of the study, it would be more accurate to make definitions such as "may be".

Reviewer #2: This article does not meet PLOS Mental Health's publication criteria. The flow between paragraphs in the introduction section was not fully achieved. The tables in the material method section are not arranged according to APA format. At the same time, the tables do not clearly present the information. The article as a draft was not considered technically sound.

6. PLOS authors have the option to publish the peer review history of their article (what does this mean? ). If published, this will include your full peer review and any attached files.

**Do you want your identity to be public for this peer review?** For information about this choice, including consent withdrawal, please see our Privacy Policy .

Reviewer #1: **Yes: ** Ziya Erokay METİN

Reviewer #2: No

---

## [Decision Letter · Decision Letter 1]

6 Nov 2024

PMEN-D-24-00001R1

Cognitive Impairment & Reporting of Hypertension among Older Adults in India: Evidence from a Population-based Study

PLOS Mental Health

Dear Dr. Dixit,

Thank you for submitting your manuscript to PLOS Mental Health. After careful consideration, we feel that it has merit but does not fully meet PLOS Mental Health’s publication criteria as it currently stands. Therefore, we invite you to submit a revised version of the manuscript that addresses the points raised during the review process.

We look forward to receiving your revised manuscript.

Kind regards,

Barkın Köse

Academic Editor

PLOS Mental Health

Journal Requirements:

Additional Editor Comments (if provided):

Reviewers' comments:

Reviewer's Responses to Questions

**Comments to the Author**

1. If the authors have adequately addressed your comments raised in a previous round of review and you feel that this manuscript is now acceptable for publication, you may indicate that here to bypass the “Comments to the Author” section, enter your conflict of interest statement in the “Confidential to Editor” section, and submit your "Accept" recommendation.

Reviewer #1: All comments have been addressed

Reviewer #2: All comments have been addressed

2. Does this manuscript meet PLOS Mental Health’s publication criteria ? Is the manuscript technically sound, and do the data support the conclusions? The manuscript must describe methodologically and ethically rigorous research with conclusions that are appropriately drawn based on the data presented.

Reviewer #1: Yes

Reviewer #2: Yes

3. Has the statistical analysis been performed appropriately and rigorously?

Reviewer #1: I don't know

Reviewer #2: Yes

4. Have the authors made all data underlying the findings in their manuscript fully available (please refer to the Data Availability Statement at the start of the manuscript PDF file)?

Reviewer #1: Yes

Reviewer #2: Yes

5. Is the manuscript presented in an intelligible fashion and written in standard English?

Reviewer #1: Yes

Reviewer #2: Yes

6. Review Comments to the Author

Reviewer #1: (No Response)

Reviewer #2: 1. The abbreviation LASI in the keywords section should be written.

2. In line 96 ‘Previous research has examined the discrepancy between self-reported hypertension and objective evaluations of hypertension evaluations and between false-negative and false-positive reporting factors and found.’ references should be added to the sentence.

3. It is said that there is an older adults group, but how were the age categories created in the characteristic features in table 1? Why are 45-65 and under 45 included?

7. PLOS authors have the option to publish the peer review history of their article (what does this mean? ). If published, this will include your full peer review and any attached files.

**Do you want your identity to be public for this peer review?** For information about this choice, including consent withdrawal, please see our Privacy Policy .

Reviewer #1: **Yes: ** Ziya Erokay Metin

Reviewer #2: **Yes: ** Kübra Ersoy

---

## [Decision Letter · Decision Letter 2]

10 Mar 2025

PMEN-D-24-00001R2

Cognitive Impairment and Reporting of Hypertension among  Adults in India: Evidence from a Population-Based Study

PLOS Mental Health

Dear Dr. Dixit,

Thank you for submitting your manuscript to PLOS Mental Health. After careful consideration, we feel that it has merit but does not fully meet PLOS Mental Health’s publication criteria as it currently stands. Therefore, we invite you to submit a revised version of the manuscript that addresses the points raised during the review process.

Please attend to the reviewer's minor concerns regarding greater clarification in the methodology and analyses.

We look forward to receiving your revised manuscript.

Kind regards,

Avanti Dey, PHD

Staff Editor

PLOS Mental Health

Journal Requirements:

Additional Editor Comments (if provided):

Reviewers' comments:

Reviewer's Responses to Questions

**Comments to the Author**

1. If the authors have adequately addressed your comments raised in a previous round of review and you feel that this manuscript is now acceptable for publication, you may indicate that here to bypass the “Comments to the Author” section, enter your conflict of interest statement in the “Confidential to Editor” section, and submit your "Accept" recommendation.

Reviewer #3: (No Response)

2. Does this manuscript meet PLOS Mental Health’s publication criteria ? Is the manuscript technically sound, and do the data support the conclusions? The manuscript must describe methodologically and ethically rigorous research with conclusions that are appropriately drawn based on the data presented.

Reviewer #3: Yes

3. Has the statistical analysis been performed appropriately and rigorously?

Reviewer #3: (No Response)

4. Have the authors made all data underlying the findings in their manuscript fully available (please refer to the Data Availability Statement at the start of the manuscript PDF file)?

Reviewer #3: (No Response)

5. Is the manuscript presented in an intelligible fashion and written in standard English?

Reviewer #3: (No Response)

6. Review Comments to the Author

Reviewer #3: Dear Authors,

The manuscript is well written. However, some changes to get a better clarity of data is needed. Please see the attached file for more details.

Additional notes:

Line 114: Why restrict only to hypertension? Why not include type 2 diabetes and other chronic medical conditions

Line 126: A more detailed description is needed for this variable as it is important. Why was the cutoff chosen as the 10th percentile, why not the 25th percentile? Do the results change if the cutoff is changed?

All the best

7. PLOS authors have the option to publish the peer review history of their article (what does this mean? ). If published, this will include your full peer review and any attached files.

**Do you want your identity to be public for this peer review?** For information about this choice, including consent withdrawal, please see our Privacy Policy .

Reviewer #3: **Yes: ** Abhishek M L

---

## [Editor Report · Decision Letter 3]

21 Apr 2025

Cognitive Impairment and Reporting of Hypertension among  Adults in India: Evidence from a Population-Based Study

PMEN-D-24-00001R3

Dear Dr. Dixit,

We are pleased to inform you that your manuscript 'Cognitive Impairment and Reporting of Hypertension among  Adults in India: Evidence from a Population-Based Study' has been provisionally accepted for publication in PLOS Mental Health.

Best regards,

Karli Montague-Cardoso

Staff Editor

PLOS Mental Health